# 4D MIND READING

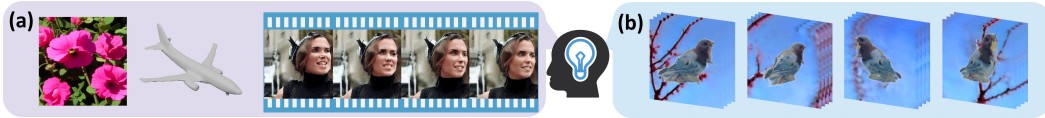

Figure 1: **fMRI signals based BCI. (a)** Subject to respective visual stimuli, prior fMRI to image, to 3D shape, and to video functions *cannot* support continuous, immersive user experience. **(b)** By generating dynamic 3D scenes from fMRI, our Brain-to-4D enables brain-driven virtual reality.

## ABSTRACT

Brain-computer interfaces (BCI) have enabled breakthroughs like translating fMRI signals into images or videos. However, human perception operates in a dynamic 3D world, processing information across both spatial and temporal dimensions. In this work, we introduce *4D Mind Reading*, a novel BCI function that generates 4D visuals—combining video and 3D structures—directly from fMRI signals. Building such a system is challenging, as training a model to generate 4D scenes from fMRI data requires paired fMRI–4D mappings, which are infeasible due to the instantaneous nature of brain responses that prevent simultaneous capture of multi-view stimuli. To address this, we propose *Mind4D*, an innovative brain-inspired fMRI conditioned 4D generation framework capable of learning asymmetric hierarchical representations from fMRI signals in a weakly supervised manner. Our approach captures both high-level and low-level representations, along with the decomposition of scene backgrounds and object foregrounds. By conditioning and integrating multiple generative priors for the foreground and background, Mind4D produces high-quality semantic 4D visuals. Extensive experiments show that Mind4D generates immersive 4D visuals semantically aligned with brain activity. Even when constrained to the reference view—the view the subject watched—our model outperforms the best fMRI-to-video approaches in CLIP-T and SSIM, achieving a 50% improvement in ICS-50 for semantic classification. We further highlight Mind4D's potential in advancing neuroscience and clinical diagnosis. Our source code will be released.

## 1 INTRODUCTION

Brain-computer interfaces (BCIs) (Saha et al., 2021; Rashid et al., 2020; Wolpaw et al., 2002) are increasingly recognized for enabling direct communication through brain activity. Among non-invasive approaches, functional magnetic resonance imaging (fMRI) has been widely adopted for various BCI functions. With recent generative AI advances, state-of-the-art fMRI decoding methods now reconstruct images (Takagi & Nishimoto, 2023; Scotti et al., 2023), videos (Chen et al., 2024; Gong et al., 2025a), and 3D shapes (Gao et al., 2023) (see Figure 1**(a)**).

However, humans possess an innate ability to perceive and interpret scenes under both spatial and temporal dimensions, even during fleeting thoughts (Heft, 2010; Kiverstein & Rietveld, 2021; Wang & Spelke, 2002). Immersed in a dynamic three-dimensional environment, the human brain inherently encodes and integrates spatiotemporal information at a subconscious level (Zhang et al., 2020; Schonhaut et al., 2023; Howard, 2017; Xu et al., 2021). Inspired by this process, we propose, for the first time, a more powerful BCI function—**4D Mind Reading**—that decodes fMRI signals into 4D visuals, integrating both video and 3D components (Figure 1**(b)**). This novel function opens up new avenues in spatiomotion-related neuroscience and interactive brain health diagnosis.

Building a 4D mind reading system is inherently challenging due to the instantaneous nature of brain responses, which prevents simultaneous multi-view capture, and the high cost and temporal

limitations of fMRI acquisition, which make it difficult to collect continuous fMRI-visual pairs. To overcome these challenges, we propose **Mind4D**, a novel brain-inspired framework for fMRI-conditioned 4D generation. Mind4D designs a novel weakly supervised learning approach that circumvents the need for direct fMRI-4D pairings. Instead, it exploits existing fMRI-2D pairs to extract asymmetric hierarchical representations from fMRI signals. These representations capture both high-level semantic information and low-level visual details, while also decomposing the scene into distinct background and foreground components. By conditioning foreground and background generative priors, Mind4D synthesizes high-quality, semantically aligned 4D visuals.

Our **contributions** are three-fold: **(i)** We introduce 4D Mind Reading, a novel BCI function that reconstructs immersive 4D visuals directly from fMRI signals. **(ii)** We propose Mind4D, an innovative brain-inspired framework that integrates asymmetric hierarchical representation learning with generative priors for fMRI-conditioned 4D synthesis. **(iii)** We establish new benchmarks for 4D mind reading, demonstrating through extensive experiments that Mind4D generates multi-view-consistent 4D visuals that are semantically aligned with brain activity. We further highlight its potential for advancing neuroscience and clinical diagnostics.

## 2 RELATED WORK

**Neural decoding for BCIs** Existing neural decoding studies (Liu et al., 2025a) have focused on extracting essential representations (Saha et al., 2021; Rashid et al., 2020) of brain signals for tasks like visual content decoding (Lawhern et al., 2018; Guger et al., 2024) and object recognition (Abdulkader et al., 2015). However, they often struggle to create detailed visuals directly from brain signals. These investigations have also facilitated advancements in reconstructing images (Beliy et al., 2019; Chen et al., 2023; Huo et al., 2024), videos (Chen et al., 2024; Lu et al., 2025; Liu et al., 2025b), and geometry (Gao et al., 2023; 2024) from fMRI data using techniques such as self-supervised dual-network architectures (Beliy et al., 2019), latent-space diffusion (Scotti et al., 2023; 2024; Gong et al., 2025a) or hierarchical latent variable models aligned with cortical processing (Takagi & Nishimoto, 2023; Miliotou et al., 2023). Co-current works explore cross-subject alignment (Wang et al., 2024; Gong et al., 2025b; Li et al., 2024), hierarchical latent decomposition (Li et al., 2025), cortical-inspired decoding (Wang et al., 2025), and multi-shot reconstruction with LLM priors (Jiang et al., 2024a). However, all previous reconstructions are limited to single-view or static objects, which pose severe limitations on immersive user experience under BCIs. Meanwhile, humans inherently encode and integrate spatial and temporal information subconsciously (Schonhaut et al., 2023; Howard, 2017; Xu et al., 2021; Rolls & Treves, 2011; Naselaris et al., 2011). We thus propose Mind4D for more seamless interaction, providing a significant step forward for applications.

**3D/4D generation** Recent advancements in text/image-based 3D generation (Poole et al., 2023; Lin et al., 2023; Wang et al., 2023; Tang et al., 2024; Liu et al., 2023; Shi et al., 2023) are predominantly based on strong 3D representations, including NeRF (Mildenhall et al., 2020), DMTet (Shen et al., 2021) or Gaussian splatting (Kerbl et al., 2023), which leverage score distillation sampling (Poole et al., 2023) (SDS) and extensive 3D datasets (Deitke et al., 2023; Yu et al., 2023; Wu et al., 2023). With the emergence of 4D representations (Wu et al., 2024; Pumarola et al., 2020; Cao & Johnson, 2023; Yang et al., 2024; 2023), these techniques have also been extended to generate dynamic 3D scenes (Jiang et al., 2024b; Ren et al., 2023; Tang et al., 2024). Our approach takes a step further by leveraging diffusion models as generative priors, which will be guided through hierarchical representations to bridge the gap between fMRI and 4D generation, highlighting its superiority in generating immersive and accurate 4D scenes from neurological data.

## 3 METHOD

### 3.1 PRELIMINARY

**Deformable 3D Gaussian splatting** 3D Gaussian splatting (3DGS) (Kerbl et al., 2023) represents a 3D scene with a set of Gaussians. Each Gaussian is characterized by position mean $\mu \in \mathbb{R}^3$, covariance matrix $\Sigma \in \mathbb{R}^{3 \times 3}$, color $\mathbf{c} \in \mathbb{R}^3$, and opacity $\alpha \in \mathbb{R}$. The color of each pixel results from the 2D projection of these 3D Gaussians and depth volumetric rendering. In the dynamic setting, we adopt deformable 3DGS (Wu et al., 2024) as the 4DGS representation, which uses an additional

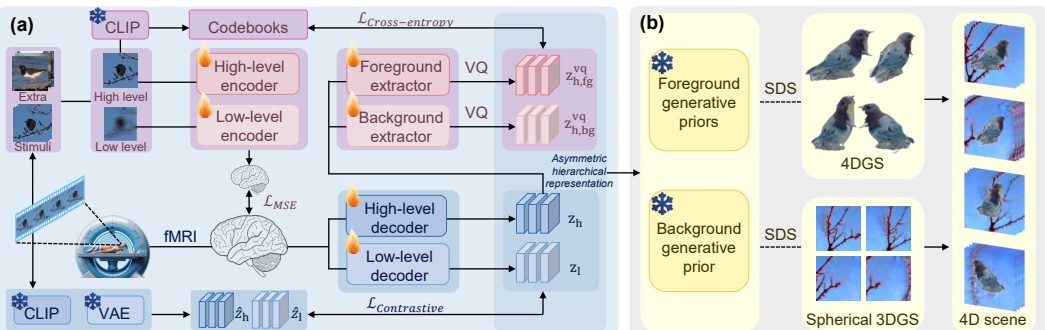

Figure 2: **Overview of Mind4D.** Without labeled fMRI-to-4D training data, we design a brain-inspired framework that integrates asymmetric hierarchical representation learning with rich generative priors for fMRI-conditioned 4D synthesis. **(a)** In weakly-supervised pre-training in the first stage, we decode fMRI into both high-level ($z_h$) and low-level ($z_l$) representations, optimized in a contrastive manner. We design fMRI encoders that can translate extra images to fMRI signals to improve the generalization of hierarchical representations. Asymmetrically, we extract forgeground ($z_{h,fg}^{vq}$) and backbround ($z_{h,bg}^{vq}$) representations from $z_h$, leveraging vector quantization (VQ) to ensure the stability of hierarchical representations. **(b)** In the second inference stage, we apply pre-trained model to extract hierarchical representation $\{z_h, z_l, z_{h,fg}^{vq}, z_{h,bg}^{vq}\}$ directly from fMRI, which conditions both foreground and background generative priors to optimize 4DGS for foreground and spherical 3DGS for background, which are then integrated for the target 4D scene composition.

network $\Phi$ to predict the deformation of $S = \{\mu, \Sigma, \alpha\}$ given timestamp $\tau$: $\tilde{S} = \Phi(S, \tau)$, where $\tilde{S}$ denotes the deformed attributes of $S$. With these deformed attributes, we can render images at different timestamps.

**Score distillation sampling** Score distillation sampling (SDS) provides a method to distill knowledge from a pre-trained diffusion model $\epsilon_\phi$. Specifically, when an image $I$ is rendered from a scene representation (*e.g.* 3DGS) parameterized by $\theta$, the gradient of SDS is calculated as:

$$\nabla_\theta \mathcal{L}_{\text{SDS}}(\phi, I_t) = \mathbb{E}\left[w(t)\left(\epsilon_\phi(I_t; t, c) - \epsilon\right)\frac{\partial I_t}{\partial \theta}\right], \tag{1}$$

where $w(t)$ is defined as the weighting function in SDS as defined in Poole et al. (2023). $I_t$ is the perturbed image with noise $\epsilon$ at time step $t$, and $c$ is the condition (*e.g.* text or image).

**Vector quantization** Vector quantization (VQ) involves mapping continuous input embeddings to discrete codebook entries. Given an input embedding $z_h \in \mathbb{R}^D$, the quantized embedding $z_h^{vq}$ is determined by selecting the closest codebook vector from a set of codebook entries $\{g_j \in \mathbb{R}^D\}_{j=1}^K$ based on $z_h^{vq} = g_k$, where $k = \text{argmin}_j \|z_h - g_j\|$.

### 3.2 Mind4D

We present **Mind4D**, a pioneer brain-inspired fMRI conditioned 4D generation framework, as depicted in Figure 2. The key idea of Mind4D is to extract asymmetric hierarchical representations from fMRI signals, replicating the intricate process of sensory decoding and encoding in human brains (Xu et al., 2021; Rolls & Treves, 2011; Naselaris et al., 2011), which condition generative priors for 4D synthesis. This approach is designed to tackle the challenge of partially aligned modalities between 2D video supervision and 4D scene targets, circumventing the need for paired fMRI-4D data. During weakly supervised pre-training in the first stage, we optimize encoders, decoders, and extractors to extract asymmetric hierarchical representations from brain signals using fMRI-2D training set (Section 3.2.1). During inference, we apply pre-trained model to extract asymmetric hierarchical representations from solely brain signals in the testing set. These hierarchical representations serve as conditions for separate generative priors for foreground and background. The resulting foreground and background 4D visuals are then integrated for 4D synthesis (Section 3.2.2).

### 3.2.1 HIERARCHICAL REPRESENTATION LEARNING

**fMRI decoding** Inspired by the human brain's ability to dynamically decode fleeting sensory inputs across spatial and temporal dimensions (Heft, 2010; Kiverstein & Rietveld, 2021; Wang & Spelke, 2002), we design fMRI decoding to decode hierarchical representations from fleeting brain signals. When perceiving scenes, humans capture both broad outlines and intricate details (Hegdé, 2008; Petras et al., 2019; Friston, 2008). To reflect this dual-resolution processing, our model adopts a two-branch architecture that explicitly separates low-level and high-level processing, allowing specialized pathways to handle global structure and fine-grained information, respectively.

Specifically, paired fMRI-2D data contain video stimuli $\hat{V}$ and $N$ sequences of brain signals $\hat{X} = [\hat{X}_1, \hat{X}_2, ..., \hat{X}_N]$. In fMRI decoding, brain signals $\hat{X}$ are mapped into visual representations $z_{\mathrm{h}} = F_{\mathrm{d,h}}(\hat{X}), z_{\mathrm{l}} = F_{\mathrm{d,l}}(\hat{X})$ through the high-level decoder $F_{\mathrm{d,h}}$ and the low-level decoder $F_{\mathrm{d,l}}$. During training, the high-level representation $z_{\mathrm{h}}$ is supervised from CLIP (Radford et al., 2021) embedding of key frame $\hat{I}$ chosen from video $\hat{V}$ through soft contrastive regulation (Scotti et al., 2023) (SoftCLIP):

$$\mathcal{L}_{\mathrm{D,H}} = \mathcal{L}_{\mathrm{SoftCLIP}}(z_{\mathrm{h}}, \hat{z}_{\mathrm{h}}), \text{ where } \hat{z}_{\mathrm{h}} = \mathrm{CLIP}(\hat{I}), \hat{I} \in \hat{V}. \tag{2}$$

On the other hand, the low-level representation $z_{\mathrm{l}}$ is supervised from SD VAE (Rombach et al., 2022) embedding of $\hat{V}$ through contrastive regulation, with

$$\mathcal{L}_{\mathrm{D,L}} = \mathcal{L}_{\mathrm{Contrastive}}(z_{\mathrm{l}}, \hat{z}_{\mathrm{l}}), \text{ where } \hat{z}_{\mathrm{l}} = \mathrm{VAE}(\hat{V}). \tag{3}$$

The high-level representation captures features of objects and scenes from key frames. The low-level representation captures rough motion by mapping the entire rough video into VAE representations, which inherently preserves essential temporal and motion information.

**fMRI encoding** The human brain's perception process involves an intricate interplay of two mechanisms: decoding sensory information from the environment and encoding meaningful representations (Xiao et al., 2024; Schmidgall et al., 2024; Xu et al., 2021; Rolls & Treves, 2011; Naselaris et al., 2011). Inspired by this, we design the fMRI encoding process in parallel with the decoding process, closely emulating this neural interplay.. The previous fMRI decoding process converts brain signals into visual representations, while the fMRI encoding process encodes visuals into brain signals. This design reflects the dual function processing of the human brain, enabling Mind4D to extract more generalized hierarchical representations from brain signals.

For the ground truth image $\hat{I} \in \hat{V}$ sampled from training videos and their Gaussian blurred representation $\hat{I}' = \mathrm{Blur}(\hat{I})$, we employ both high-level encoder $F_{\mathrm{e,h}}$ and low-level encoder $F_{\mathrm{e,l}}$ to encode images into brain signals: $X_{\mathrm{h}} = F_{\mathrm{e,h}}(\hat{I}), X_{\mathrm{l}} = F_{\mathrm{e,l}}(\hat{I}')$. These encoders are supervised through Mean Square Error (MSE):

$$\mathcal{L}_{\mathrm{E,H}} = \mathcal{L}_{\mathrm{MSE}}(X_{\mathrm{h}}, \hat{X}), \ \mathcal{L}_{\mathrm{E,L}} = \mathcal{L}_{\mathrm{MSE}}(X_{\mathrm{l}}, \hat{X}). \tag{4}$$

More importantly, for external images $\hat{I}_{extra}$ that do not have corresponding fMRI pairs, the design of the encoders could model surrogate brain signals $X_{extra} = F_{\mathrm{e,h}}(\hat{I}_{extra})$. These surrogate fMRI-visual pairs are subsequently fed into decoders $F_{\mathrm{d,h}}, F_{\mathrm{d,l}}$ as augmentation, simulating the human brain's continuous perception of different layouts from the external world to enhance the generalization of our weakly supervised representation learning framework.

**Vector quantized semantic extraction** Inspired by the role of Medial Temporal Lobe (MTL) in the interpretation of high-level information (Squire et al., 2004; Eichenbaum et al., 2007), we further design an asymmetric representation learning to extract semantic representation solely from high level representations $z_{\mathrm{l}}$ in a vector quantized process. This process emulates the ability of MTL to improve representation learning by capturing higher-level abstractions in visual and scene data.

Specifically, we design two extractors $F_{\mathrm{fg}}, F_{\mathrm{bg}}$ to split the high-level representation $z_{\mathrm{h}} = F_{\mathrm{d,h}}(\hat{X})$ into representations that represent background and foreground information: $z_{\mathrm{h,fg}} = F_{\mathrm{fg}}(z_{\mathrm{h}}), z_{\mathrm{h,bg}} = F_{\mathrm{bg}}(z_{\mathrm{h}})$. The resulting $z_{\mathrm{h,fg}}, z_{\mathrm{h,bg}}$ is then vector quantized by:

$$\begin{aligned} z_{\mathrm{h,fg}}^{vq} &= g_{\mathrm{fg}}[k], \text{ with } k = \mathrm{argmin}_j \|z_{\mathrm{h,fg}} - g_{\mathrm{fg}}[j]\|, \\ z_{\mathrm{h,bg}}^{vq} &= g_{\mathrm{bg}}[k], \text{ with } k = \mathrm{argmin}_j \|z_{\mathrm{h,bg}} - g_{\mathrm{bg}}[j]\|, \end{aligned} \tag{5}$$

where $g_{\text{fg}} \in \mathbb{R}^{K_{\text{fg}} \times D_{\text{fg}}}, g_{\text{bg}} \in \mathbb{R}^{K_{\text{bg}} \times D_{\text{bg}}}$ are the codebooks with $K_{\text{fg/bg}}$ entries and $D_{\text{fg/bg}}$ dimension. They are obtained by mapping the video stimuli $\hat{V}$ into the CLIP (Radford et al., 2021) space through our prelabeled foreground and background annotations. The extractors $F_{\text{fg}}, F_{\text{bg}}$ are optimized using cross-entropy (CE) loss as equation (6):

$$\mathcal{L}_{\text{VQ}} = \mathcal{L}_{\text{CE}}(z_{\text{h,fg}}, g_{\text{fg}}) + \mathcal{L}_{\text{CE}}(z_{\text{h,bg}}, g_{\text{bg}}). \tag{6}$$

One key advantage of vector quantization is its ability to bypass the curse of dimensionality. By constraining the size of the latent space $K \ll n$, we significantly improve the regularization of the model and avoid overfitting (Peng et al., 2023) in high-dimensional feature spaces. Furthermore, our approach significantly reduces the KL divergence between empirical and ground truth distributions, as indicated by Theorem 3.1. It shows that the quantized latent space representation $z_{\text{h}}^{vq}$ (either $z_{\text{h,fg}}^{vq}$ or $z_{\text{h,bg}}^{vq}$) yields a much tighter approximation to the true distribution compared to the nonquantized representation $z_{\text{h}}$, which is crucial for robust latent representations.

**Theorem 3.1.** *Denote $p(z_{\text{h}})$ as the distribution of the embeddings without vector quantization and $p(\hat{z}_{\text{h}})$ as the smooth-approximated empirical distribution from samples, $p(z_{\text{h}}^{vq})$ and $p(\hat{z}_{\text{h}}^{vq})$ as their vector quantized counterparts. Then,*

$$KL(p(z_{\text{h}}^{vq})||p(\hat{z}_{\text{h}}^{vq})) \ll KL(p(z_{\text{h}})||p(\hat{z}_{\text{h}})). \tag{7}$$

Additionally, Theorem 3.2 shows our vector quantized approach also significantly reduces entropy. This ensures that the model is less likely to capture irrelevant data-specific noise, thereby enhancing generalization to unseen data.

**Theorem 3.2.** *Denote $L$ as the CLIP (Radford et al., 2021) space boundary size, $H(z_{\text{h}}^{vq})$ as the entropy of distribution of vector quantized embeddings, and $H(z_{\text{h}})$ as the entropy of Riemann-Discrete approximated distribution without vector quantization. Then we have $H(z_{\text{h}}) > H(z_{\text{h}}^{vq})$,*

$$H(z_{\text{h}}) - H(z_{\text{h}}^{vq}) = O\left(\log\left(L^d/K\right)\right). \tag{8}$$

The proof can be found in Section F and Section G in the supplementary. The empirical validation and ablation in Section D.4 illustrate the substantive role of the quantized latent space representation. As a result, the whole loss in our training stage is:

$$\mathcal{L} = \lambda_{\text{E}}(\mathcal{L}_{\text{E,H}} + \mathcal{L}_{\text{E,L}}) + \lambda_{\text{D}}(\mathcal{L}_{\text{D,H}} + \mathcal{L}_{\text{D,L}}) + \lambda_{\text{VQ}}\mathcal{L}_{VQ}. \tag{9}$$

Different from previous weakly supervised approaches (Zhou, 2018; Mahajan et al., 2018; Zheng et al., 2021) that assume uni-modality labels are available, our framework solves the partially aligned modalities between existing fMRI-2D pairs and fMRI-4D targets. This is achieved by extracting asymmetric hierarchical representations from the fMRI-2D supervision and serving these hierarchical representations as the condition for the 4D scene targets.

### 3.2.2 GENERATIVE PRIORS FOR FMRI-BASED 4D SYNTHESIS

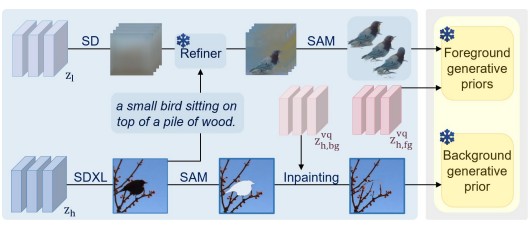

Figure 3: **Hierarchical representation guidance.** The hierarchical representations $z_l, z_h$ generate both coarse videos and fine key-frame visuals, which are then refined and segmented to get foreground visuals to condition the foreground generative priors with $z_{\text{h,fg}}^{vq}$. The $z_{\text{h,bg}}^{vq}$ inpaints the segmented key-frame background, conditioning the background generative prior.

**Hierarchical representation guidance** After our weakly supervised pretraining, we apply pre-trained decoders $F_{\text{d,h}}, F_{\text{d,l}}$, and extractors $F_{\text{fg}}, F_{\text{bg}}$ to extract asymmetric hierarchical representations $z_h, z_l, z_{\text{h,fg}}^{vq}, z_{\text{h,bg}}^{vq}$ from soly fMRI in the testing set. These hierarchical representations are further pre-processed to guide generative priors for 4D synthesis, depicted in Figure 3. The high-level representation $z_h$ is used to generate key-frame visuals $I_h = \text{SDXL}(z_h)$ through SDXL (Podell et al., 2023), while the low-level representation $z_l$ is applied to generate low-level videos $V_l = \text{VAE-D}(z_l)$ frame by frame through the VAE decoder of Stable Diffusion (Rombach et al., 2022). The key-frame visuals $I_h$, low-level videos $V_l$ and the caption generated from key-frame visuals $c_h =$

BLIP($I_{\text{h}}$) are integrated together to generate refined video $V_{\text{R}}$ through a pretrained refiner $F_{\text{r}}$:$V_{\text{r}} = F_{\text{r}}(V_l; c_{\text{h}}, I_{\text{h}})$. The refined video $V_{\text{r}}$ is further segmented into foreground part through SAM (Kirillov et al., 2023). For the background part, we remove the objects in key-frame visuals $I_{\text{h}}$ and inpaint it using $z^{vq}_{\text{h,bg}}$ as semantic guidance.

**Foreground and background generative priors** Modeling 4D scenes faces two challenges: (1) Foreground and background present intrinsically different characteristics (*e.g.*, dynamic vs. static); (2) Camera perspectives in 4D scenes often blur out nearby objects dynamically. To tackle these issues, we propose decomposed generation with both foreground and background generative priors.

The foreground generative priors are aimed at optimizing 4DGS (*i.e.*, deformable 3D Gaussians introduced in Section 3.1), driven by foreground video $V_{\text{r}}$ and semantic representation $z^{vq}_{\text{h,fg}}$. The foreground optimization, 3D Gaussians and its deformation are optimized by foreground generative priors using SDS loss as equation (1). Along with a $L_2$ loss between rendered image $\hat{I}_{\text{ref}}$ under reference views with $I_{\text{ref}} \in V_{\text{r}}$, the total loss $\mathcal{L}_{\text{fg}}$ for foreground modeling can be expressed by:

$$\mathcal{L}_{\text{fg}} = \lambda_{\text{img}}\mathcal{L}_{\text{SDS,img}} + \lambda_{\text{sem}}\mathcal{L}_{\text{SDS,sem}} + \lambda_{\text{ref}}\|\hat{I}_{\text{ref}} - I_{\text{ref}}\|_2^2, \tag{10}$$

where $\lambda_*$ are balancing weights, with image-based generative prior ("img") and semantic-based generative prior ("sem") referring to AI (2023) and Shi et al. (2023) guidance, respectively.

The background generative prior optimizes spherical 3D Gaussians whose points are scattered on a spherical surface. This spherical 3DGS is designed to mimic a panoramic environment map, which supports rendering from $360°$ viewpoints. A scene-level generative prior (Sargent et al., 2023) extends the inpainted background image $I_{\text{h}}$ into a $360°$ environment. The background loss $\mathcal{L}_{\text{bg}}$ is:

$$\mathcal{L}_{\text{bg}} = \lambda_{\text{bg}}\mathcal{L}_{\text{SDS,bg}} + \lambda_{\text{ref}}\|\hat{I}_{\text{h}} - I_{\text{h}}\|_2^2, \tag{11}$$

where $\lambda_*$ refers to balancing weights and $\mathcal{L}_{\text{SDS,bg}}$ to SDS under background generative prior.

**Integration** Finally, we integrate the separately optimized foreground and background. To obtain the composite image $I_c$, we render both foreground image $I_{\text{fg}}$ and background image $I_{\text{bg}}$ with a foreground mask $M$, and then blend them by: $I_c = I_{\text{fg}} \odot M + I_{\text{bg}} \odot (1 - M)$. At this stage, we design to use the another fMRI-to-video generative prior (Chen et al., 2024) to directly produce the reference video $\{I_{\text{rv}_k}\}_{k=1}^T$ by denoising a noise-perturbed composite video $\{I_{c_k}\}_{k=1}^T$. A simple $L_2$ loss between $I_{c_k}$ and $I_{\text{rv}_k}$ is applied to train both 4DGS and spherical 3DGS.

### 3.3 Neuroscience interpretability and diagnosis

We apply Mind4D to two key applications: neuroscience interpretability and diagnosis (Figure 5). Our design focuses on four specific groups within the visual cortex: primary (V1), associative (V2, V3, V4), dynamic (MT, MST, LIP), and synthesis (TPOJ) visual cortex. For each group, we examine the role by masking out other region groups and decoding each region of interest (ROI) group separately. To simulate disorder diagnosis, we introduce perturbations to each group and analyze the resulting 4D scenes to evaluate their functional impact.

## 4 Experiments

**Dataset** Our research extends publicly available fMRI-video dataset (Wen et al., 2018). The fMRI data are acquired using a 3T MRI scanner at a repetition time (TR) of 2 seconds, comprising 18 segments of 8-minute video clips, resulting in 4,320 training video-fMRI pairs, and 5 segments for 1,200 testing samples. For each video-fMRI pair, a single frame is randomly selected as the ground truth image for background supervision. Besides, we annotated the video-fMRI samples with foreground objects (Krizhevsky et al., 2009) and background scenes (Bansal, 2019). Lacking 4D annotations, we employ semantic embeddings of these labels as a codebook to supervise our foreground and background extractors.

**Metrics** In line with Chen et al. (2024), we employ the Structural Similarity Index Measure (SSIM) for pixel-level accuracy and classification-based score for semantic accuracy with respect to ground truth visual stimuli. The classification score compares the top-1 accuracy between ground truth and rendered frames across selected $N = 2$ and $N = 50$ classes, with 100 repetitions for an

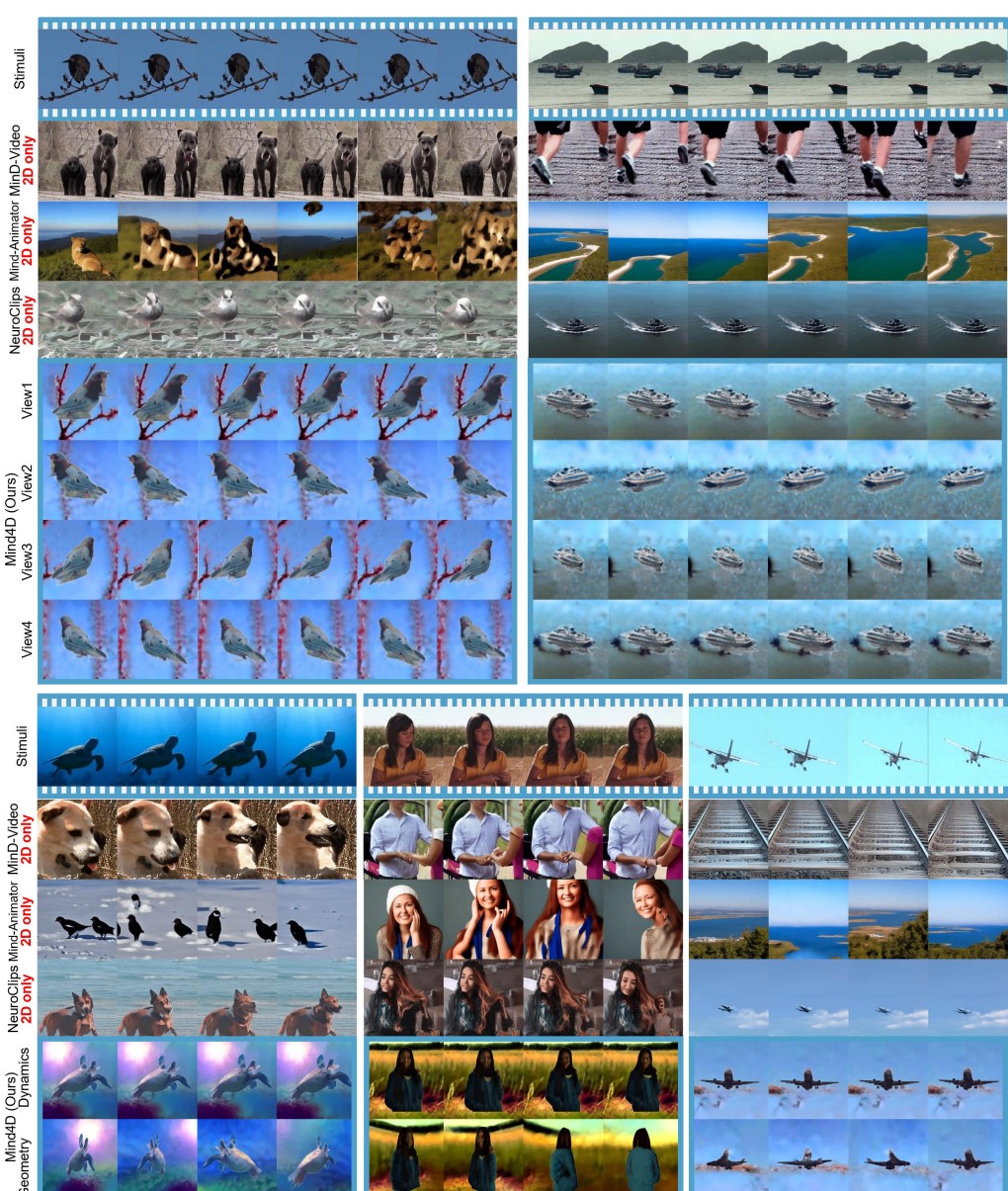

Figure 4: **Multi-view 4D scenarios of Mind4D**. Previous methods (MinD-Video (Chen et al., 2024), Mind-Animator (Lu et al., 2025) and NeuroClips (Gong et al., 2025a)) are limited in **2D** and the reconstruction results usually do not align well with visual stimuli. In comparison, Mind4D not only achieves higher consistency with visual stimuli, but also pioneers the **4D Mind Reading** function through a novel weakly supervised framework. Please refer to the supplementary material for additional comparisons and visualizations and the accompanying video for dynamic results.

average success rate and standard deviation. Both image and video classifiers are used, designed as ICS-$N$ and VCS-$N$, respectively. Additionally, following Yin et al. (2023); Pan et al. (2024b), we incorporate CLIP-T as a 4D metric, which evaluates the temporal smoothness by computing the CLIP similarity between adjacent frames in a rendered video. Except for reporting CLIP-T of videos at specific views in Yin et al. (2023); Pan et al. (2024b), we also adopt a 360° video around the 4D scene which represents the spatial geometry, resulting in CLIP-T-G. To further inspect 3D geometric consistency, we conduct a user study by showing videos to 7 participants and asking them to judge how well the contents present 3D/4D geometry. The success rate (SR, ratio of "yes" answers) is used as the geometry metric. For 4D benchmark, we render a 4D model from the front view (reference view), side views, and back view, with each view evaluated separately across 100 cases.

Table 1: **Quantitative comparison with fMRI decoding methods.** MinD-Video (Chen et al., 2024), Mind-Animator (Lu et al., 2025), NeuroClips (Gong et al., 2025a), and Neurons (Wang et al., 2025) only serve as comparisons for the front view as they lack 3D geometry. The best performance is highlighted in bold, while the second-best performance is shown with underlined text.

| Method | View | Video-based | | Frame-based | | | 4D-based | |
| --- | --- | --- | --- | --- | --- | --- | --- | --- |
| | | VCS-2 ↑ | VCS-50 ↑ | ICS-2 ↑ | ICS-50 ↑ | SSIM ↑ | CLIP-T ↑ | CLIP-T-G ↑ |
| MinD-Video | Front | 0.8545 | 0.1318 | 0.7962 | 0.2061 | 0.2466 | 0.9434 | - |
| Mind-Animator | Front | 0.8727 | 0.2189 | 0.8365 | 0.2655 | 0.3942 | 0.8516 | - |
| NeuroClips | Front | 0.8827 | **0.2291** | 0.8368 | 0.3020 | 0.3432 | 0.9409 | - |
| Neurons | Front | 0.8815 | 0.2193 | 0.8573 | 0.3251 | 0.3741 | 0.9502 | - |
| **Mind4D** **(Ours)** | Front | **0.8918** | 0.2209 | **0.8742** | **0.4532** | **0.4262** | **0.9683** | - |
| | Side | 0.8255 | 0.2009 | 0.8641 | 0.3115 | 0.4129 | 0.9717 | - |
| | Back | 0.8682 | 0.2336 | 0.8986 | 0.3736 | 0.4249 | 0.9761 | - |
| | Mean | 0.8618 | 0.2185 | 0.8790 | 0.3794 | 0.4213 | 0.9720 | 0.9380 |

Table 2: **Quantitative comparison with 3D generation methods.** We compare against Zero123 (Liu et al., 2023), MVDream (Shi et al., 2023), and ZeroNVS (Sargent et al., 2023). The video-based and frame-based results are averaged across all views. The best scores are highlighted in **bold**, and the second-best are underlined.

| Method | Video-based | | Frame-based | | | 4D-based | | |
| --- | --- | --- | --- | --- | --- | --- | --- | --- |
| | VCS-2 ↑ | VCS-50 ↑ | ICS-2 ↑ | ICS-50 ↑ | SSIM ↑ | CLIP-T ↑ | CLIP-T-G ↑ | SR ↑ |
| Zero123 | 0.8429 | 0.1787 | 0.7372 | 0.2942 | 0.3784 | 0.9531 | 0.9348 | 0.6514 |
| MVDream | 0.8374 | 0.1219 | 0.7144 | 0.1186 | 0.2919 | 0.9517 | 0.9338 | 0.5700 |
| ZeroNVS | 0.4608 | 0.0687 | 0.5668 | 0.0708 | 0.2462 | 0.9288 | 0.9307 | 0.2843 |
| **Mind4D (Ours)** | **0.8618** | **0.2185** | **0.8790** | **0.3794** | **0.4213** | **0.9720** | **0.9380** | **0.8300** |

**Implementation details** Our designed high-level encoder $F_{e,h}$, decoder $F_{d,h}$, low-level encoder $F_{e,l}$, decoder $F_{d,l}$, foreground extractor $F_{fg}$ and background extractor $F_{bg}$ are all MLP structures. The external images $\hat{I}_{extra}$ are selected from COCO (Lin et al., 2014). We set $\lambda_D = 1.0$, $\lambda_E = 0.3, \lambda_{VQ} = 0.1$ for pre-training. Foreground generative prior uses pretrained models from AI (2023); Shi et al. (2023), while the background generative prior employs Sargent et al. (2023). The fMRI-to-video generative prior exploits structures from Chen et al. (2024). We set $\lambda_{img} = 1, \lambda_{text} = 0.5, \lambda_{ref} = 10,000, \lambda_{env} = 1$. More details are in Section C in supplementary.

### 4.1 4D GENERATION RESULTS

We present our 4D generation results in Figure 4, Table 1, and Table 2. For comparison with fMRI decoding methods, we include results with MinD-Video (Chen et al., 2024), Mind-Animator (Lu et al., 2025), NeuroClips (Gong et al., 2025a), and Neurons (Wang et al., 2025). In the visual results in Figure 4, all previous methods are limited to 2D reconstructions and are frequently inconsistent with visual stimuli. In comparison, our method ensures greater consistency with visual stimuli and pioneers the groundbreaking 4D Mind Reading function. Moreover, our method achieves a higher SSIM (Wang et al., 2004) score from the reference view (front view) as detailed in Table 1. Regarding semantic-level metrics, our method achieves comparable success rates from the reference front view, achieving a 50% improvement in ICS-50 for semantic classification compared with Neuro-Clips (Gong et al., 2025a). For CLIP-T scores assessing the 4D effect, our results demonstrate both dynamic and spatial smoothness, all outperforming previous methods, which focus on single-view output. Please refer to the supplement for more visualization results and comparisons.

For comparison with 3D/4D generations, we leverage results with ZeroNVS (Sargent et al., 2023), Zero123 (Liu et al., 2023), and MVDream (Shi et al., 2023). Existing 3D/4D modeling or multi-view diffusion methods cannot be applied for fMRI-conditioned 4D generation since they do not take fMRI as input. We thus create the best possible competitors by streamlining two successive steps: (i) fMRI→video: We derive low-level video $V_l$ and text $c_h$ using the same methods described in Sections 3.2.1 and 3.2.2; (ii) Video→4D: We then use these outputs ($V_l$ or $c_h$) as direct inputs to a diffusion model to generate the final Gaussians. As illustrated in Table 2, our Mind4D con-

Figure 5: **ROI (region of interest) interpretability and diagnosis.** Our proposed Mind4D can separately encode distinct visual cortex groups for Neuroscientific research, and could conduct diagnosis on various brain regions. For each region in (a), we first assess ROI interpretability by independently applying Mind4D to each unique ROI, generating 4D synthesized representations in (b). Subsequently, we apply perturbations to individual regions and leverage Mind4D to simulate disorder-specific diagnostic analysis for 4D scenes in (c).

sistently outperforms all alternatives across visual consistency scores (VCS), identity consistency scores (ICS), and perceptual metrics (SSIM, CLIP-T, CLIP-T-G, and SR).

## 4.2 ABLATION STUDY

We conduct ablations on all components of Mind4D to show their contributions. We separately ablate on fMRI encoding (Section D.1), decomposed generation of the foreground and background (Section D.2), hierarchical representations (Section D.3), vector quantization (Section D.4) and the impact of fMRI signals (Section D.5). Ablation results show that all previous designs are crucial for fMRI extraction and 4D synthesis. Please refer to section D in the supplementary for details.

## 4.3 COMPREHENSIVE ROI ANALYSIS

**ROI interpretation** The function of each specific ROI group is also analyzed separately (Figure 5(b)). The V1 visual region maintains the initial processing of the edges, orientations, and spatial frequencies of the scene, confirming its essential role in the detection of basic visual features. The associative (V2, V3, V4) only generate certain details of faces and hairs, which cannot independently decode visuals, indicating their dependence on V1 for information processing. Meanwhile, the spatiomotion (MT, MST, LIP) regions could only generate overall motion and flow, contributing little to complex patterns and shapes. The TPOJ region includes a cohesive visual experience, illustrating its role in information integration. These findings align well with previous research on region-of-interest (ROI) functionality in visual perception (Tong, 2003; Kim et al., 2020).

**ROI diagnosis** As depicted in Figure 5(c), the disorder in the primary (V1) visual regions leads to impairments in overall visual comprehension, supporting centrality in foundational and complex visual processing. Disorders in the synthesis (TPOJ) region result in a more comprehensive disruption of scene perception, suggesting its crucial role in integrating visual inputs into a coherent whole. In contrast, the disorder in spatiomotion (MT, MST, LIP) indicates that these regions have only marginal effects on directions and motions.

## 5 CONCLUSION

In this study, we introduce Mind4D, a pioneering framework tailored for the newly proposed Mind4D BCI function, enabling the generation of dynamic 3D scenes from brain fMRI signals for immersive user experience. The core idea is the brain-inspired representation learning approach, which processes fMRI signals effectively into hierarchical representations, seamlessly serving as conditions for generative priors for 4D scene synthesis. Through this weakly supervised design, Mind4D overcomes the challenges posed by the absence of fully supervised 4D brain training data. Experimental results have demonstrated the capability of Mind4D in decoding time-continuous and view-consistent 4D visuals closely aligned with the underlying brain activity. We hope this work can open up and foster more advanced research and applications in BCI and neuroscience studies.

## 6 ETHICS STATEMENT

We believe that our proposed task and method has promising applications in Brain-Computer Interfaces. However, every method that learns from data carries the risk of introducing biases. In the fMRI encoding stage, all the encoders are trained on open-source brain datasets described in Section 4. The subsequent generation stage is based on the open-source diffusion models that are pre-trained on the data from the Internet. Therefore, work that basesd on our method should carefully consider the consequences of any potential underlying risks and biases.

## 7 REPRODUCIBILITY STATEMENT

We are committed to the reproducibility of Mind4D. We will release the full code upon the final acceptance of the paper. To facilitate verification before code release, we have thoroughly described our in Section 3 and provided comprehensive implementation details in Section C. These sections cover all relevant aspects of multi-faceted encoding and 4D scene generation.

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

## A    LARGE LANGUAGE MODELS (LLM) USAGE

We made limited use of Large Language Models (LLMs) as a writing assistant. Specifically, LLMs were utilized to correct grammar errors to improve readability. We take full responsibility for the content in this work.

## B    LIMITATIONS AND FUTURE WORKS

As a pioneering exploration of the 4D Mind Reading function, we only examined the feasibility of Mind4D on Wen et al. (2018). We plan to extend our framework to apply it to other datasets with different modalities and clinical settings in future work.

Furthermore, due to the absence of 4D ground truth, current evaluations on geometry are limited to CLIP similarity and the user study. As this study represents the very first work in 4D mind-reading, our focus is on evaluating consistency across semantics, texture, and geometry. This is achieved through multiple widely adopted metrics: VCS-2, VCS-50, ICS-2, ICS-50, SSIM, CLIP-T and CLIP-T-G. Our propose SR in user study partly evaluate geometric quality to some extent. However, more objective geometry metrics on 3D/4D fidelity, such as depth consistency, mesh normal consistency, or multi-view photometric error, remain unexplored. These metrics objective geometry metrics could be further evaluated once 4D ground truth is available.

Moreover, the overall quality of the generated 4D content is currently constrained by fMRI decoding (Gong et al., 2025a) and generation priors (AI, 2023; Sargent et al., 2023; Shi et al., 2023). These generation priors may introduce their own noise and biases for the final results. Due to the fundamental absence of 4D ground truth from brain signals, the reliance on such external knowledge is necessary in current research. While this pioneering work may not yet be sufficient for direct, immediate practical applications, we firmly believe this study opens up an exciting and highly promising direction for future research in fMRI-driven 4D content generation and its transformative potential for clinical and neuroscience fields.

Our application on spatiomotion-related neuroscience and interactive brain health diagnosis could also be further developed with improved models and clinical experiments. The other potential real-world applications for Mind4D include:

(1)  Brain-driven virtual reality for immersive communication and interaction, such as enabling users to navigate virtual spaces using only their thoughts. Advanced gaming experiences controlled by brain signals can offer new levels of immersion and interaction.

(2)  In neurorehabilitation, it can simulate realistic environments for stroke patients to practice daily activities.

(3) Brain-driven creativity allows artists to produce 3D movies and artistic expressions using only their thoughts, thus unlocking new forms of immersive artistic expression.

(4)  Educational tools can provide interactive, brain-responsive simulations, such as virtual science experiments controlled by students' brain activity.

## C    IMPLEMENTATION DETAILS

### C.1    HIERARCHICAL REPRESENTATION LEARNING

The high-level decoder $F_{d,h}$ first employs an MLP to map the fMRI data into a 1024-dimensional vector. This is followed by four MLPs with residual connections to further extract fMRI features. The output is then transformed into $257 \times 768$-dimensional shared feature representations, which employs diffusion prior for the high-level representation $Z_h$. The low-level decoder $F_{d,l}$ utilizes shallow MLP modules to extend fMRI data to match the dimensions of video frame lengths. Subsequently, a Temporal Upsampling module Gong et al. (2025a) is applied sequentially to generate the embedding $z_l$, which can then be fed into the Stable Diffusion VAE decoder. Both high-level and low-level encoding process begins with inputting image patches into a 12-layer vision transformer with a token size of 768, pre-trained using the DINOv2 Oquab et al. (2024) to extract features. In

the transformer architecture, we choose the query features, as they are known to exhibit more object-centric feature representations to extract pertinent information from the encoder's output, which is used to predict fMRI.

Both the foreground and background extractors use two-layer MLPs to map $z_h$ into $z_{h,fg}, z_{h,bg}$. The codebook dimensions for foreground modeling are set to $D_{fg} = 77 \times 1024$, aligned with Shi et al. (2023), while the background modeling follows Takagi & Nishimoto (2023) with dimensions of $D_{bg} = 77 \times 768$. Given the practical challenges in acquiring sufficient 4D stimuli for end-to-end optimization, these codebooks are crafted around specific categories of foreground objects Krizhevsky et al. (2009) and background scenes Bansal (2019). The foreground object categories refer to the category of CIFAR-100 Krizhevsky et al. (2009), while the background scene categories exploit the categories of Intel Image Classification Bansal (2019). The number of codewords of foreground and background is set in correspondence with the number of categories in these protocols.

We set $\lambda_D = 1.0$, $\lambda_E = 0.3$ and $\lambda_{VQ} = 0.1$ for training. The whole high-level encoding process is trained with 150 epochs, while the whole low-level encoding process is trained with 50 epochs. After each 10 epochs, we apply trained encoders to encode extra images from the training set of the Natural Scenes Dataset (NSD) Allen et al. (2022) to update surrogate fMRI. To encourage temporal alignment between video and fMRI signals and allow fair comparison with prior work, we downsampled the videos from 30 FPS to 3 FPS. Hence, we adopted a fixed delay for simplicity and consistency, as adopted in Gong et al. (2025a). Training all fMRI encoders, decoders, and extractors is a one-time process that takes approximately two days on one NVIDIA A6000 GPU. Once completed, the parameters are fixed for subsequent 4D generation from any fMRI.

### C.2 GENERATION

In the generation stage, we implement our pipeline based on the DreamGaussian4D (Ren et al., 2023), a framework focusing on efficient 4D generation. Training involves 500 steps for static foreground and background, 1,000 steps for dynamic foreground, and 50 steps for joint refinement. The Gaussians are initialized with 5,000 random points for foreground inside a sphere of and 200,000 random points for background around a sphere of radius 5. Densification is performed every 50 steps. For balancing weights, we set $\lambda_{img} = 1, \lambda_{text} = 0.5, \lambda_{ref} = 10,000, \lambda_{env} = 1$. All hyper-parameters are selected based on common experimental tuning, and are set to the same value for all individuals.

For diffusion guidance, we use pretrained models from Stable Zero123 (AI, 2023) and MV-Dream (Shi et al., 2023) object-level 3D-aware diffusion, adopt ZeroNVS (Sargent et al., 2023) as 2D prior in scene-level 3D-aware diffusion, and apply MinD-Video (Chen et al., 2024) for fMRI-to-video diffusion prior. Considering the unstable training of Gaussians in the generative manner, we follow (Pan et al., 2024a) to manually clip the gradient of rendered image pixel-wisely. The whole generation pipeline takes about 30 minutes on one NVIDIA A6000 GPU. Following this, the parameters for 4D Gaussian splatting are saved, enabling future inference processes. This setup allows for an inference speed of 15 frames per second (FPS), supporting real-time interaction.

## D ABLATIONS

Table 3: **Ablation on fMRI encoding.** fMRI encoding provides largely consistent gain.

| Encoding | VCS-2 | VCS-50 | ICS-2 | ICS-50 | SSIM | CLIP-T | CLIP-T-G | SR |
|---|---|---|---|---|---|---|---|---|
| w/o | 0.7938 | 0.1113 | 0.7587 | 0.1167 | 0.3535 | 0.9707 | **0.9382** | 0.7957 |
| w/ | **0.8618** | **0.2185** | **0.8790** | **0.3794** | **0.4213** | **0.9720** | 0.9380 | **0.8300** |

### D.1 FMRI ENCODING

Figure 6 and Table 3 highlights the importance of our designed fMRI encoding process, which significantly improves overall quality. This is achieved by introducing surrogate fMRI from extra data, which can enhance the generalization of our hierarchical representations.

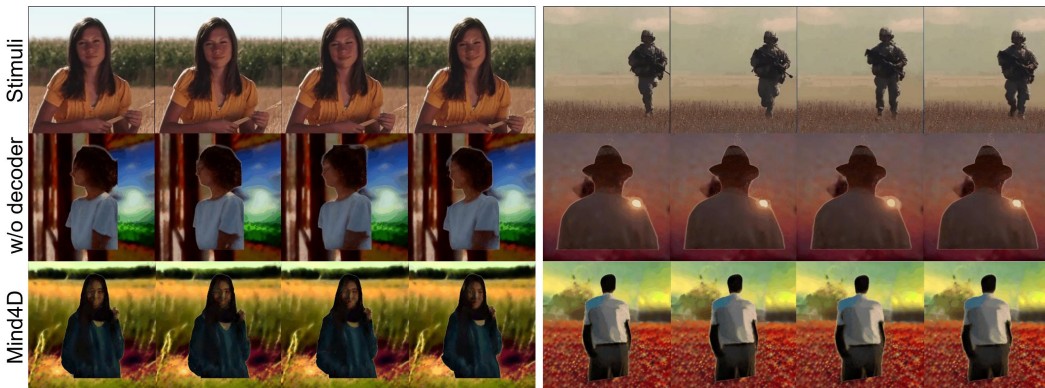

Figure 6: **Ablation on fMRI encoding.** Our design of fMRI encoders enhances the generalizability of hierarchical representations.

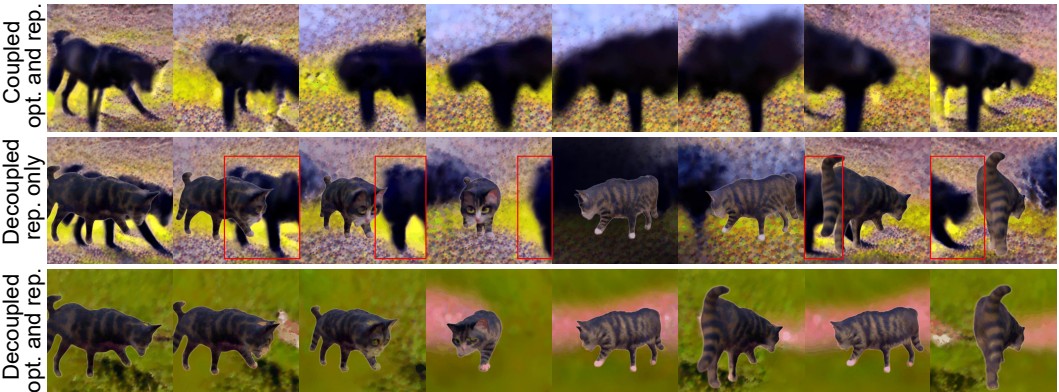

Figure 7: **Ablation on decoupling-coupling.** The "rep." denotes scene representation and the "opt." denotes scene optimization. The coupling of representations leads to bad geometry and the coupling of optimization leads to ambiguity.

### D.2 DECOMPOSED GENERATION STRATEGY

In Figure 7, we performed an ablation study on the decomposed generation of the foreground and background. Our method adopts both decoupled scene representations (*i.e.*, 4DGS and spherical 3DGS) and decoupled optimization strategy introduced in Figure 3.2.2. For ablation, we study the coupled optimization strategy, which employs composite videos as reference while simultaneously optimizing scene representations. We also investigate the coupled scene representation, which relies solely on 4DGS to jointly model foreground and background. As shown in Figure 7, we find that the coupling of foreground and background poses the challenge to the optimization of 4D scene, while the decoupled design achieves the best geometry and avoids the ambiguity between the foreground and background.

### D.3 IMPACTS OF HIERARCHICAL REPRESENTATIONS

We further investigate the impacts of hierarchical representations: semantic representation $z_{\text{h,fg}}^{vq}$ and video $V_r$ derived from representations $z_\text{h}, z_\text{l}$ on foreground generation, as shown in Figure 8. Since the reference frames are typically out of distribution of the training data (Deitke et al., 2023) used for generative prior models, the baseline "w/o sem" that relies solely on image-based generative prior (with no guidance from semantic representation $z_{\text{h,fg}}^{vq}$ and semantic-based generative prior) fails to produce satisfactory 3D shapes. In addition, the results using only semantic-based generative prior ("w/o img") do not accurately reflect the brain-related images.

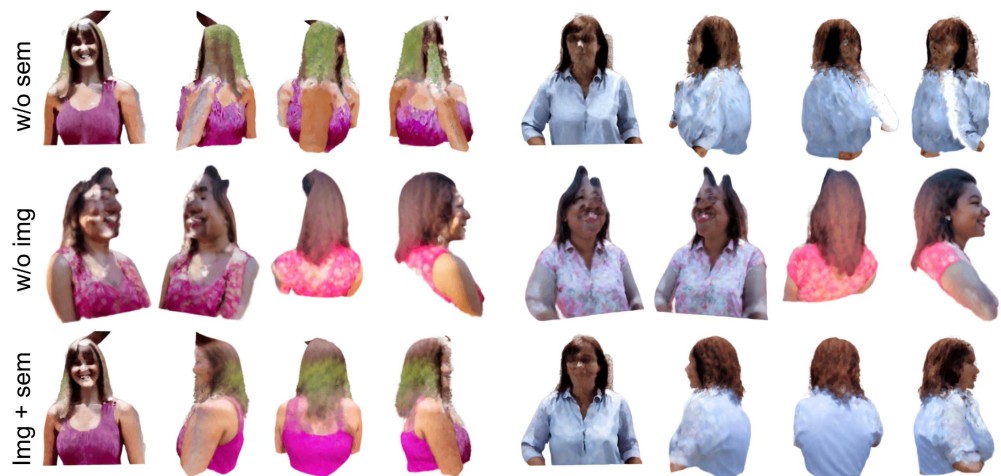

Figure 8: **Ablation on the hierarchical representations for foreground generative priors.** Without either semantic (sem) representation $z_{\text{h,fg}}^{vq}$ or guided image (img) generated from $z_{\text{h}}, z_{\text{l}}$ for 3D appearance guidance, the rendering quality decreases significantly.

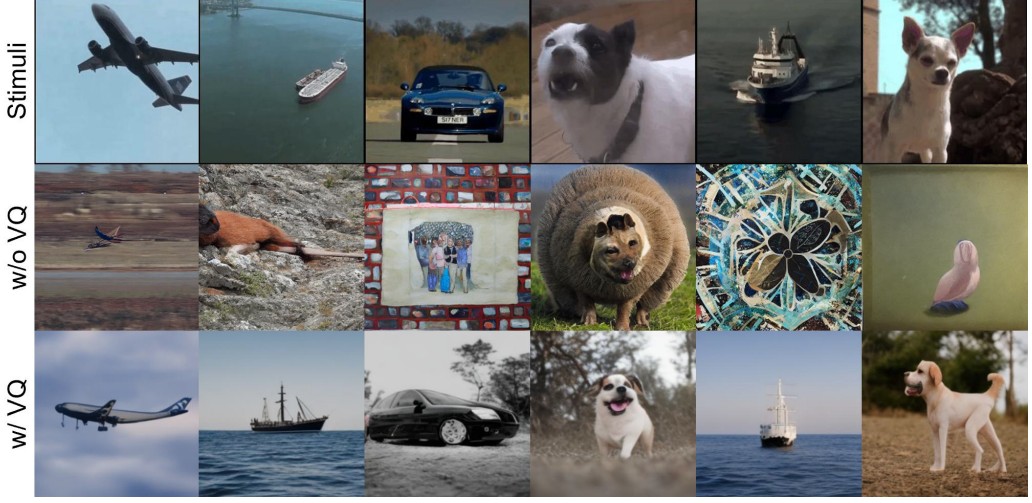

Figure 9: **Ablation on vector quantization (VQ).** VQ significantly improves the stability of hierarchical representations.

### D.4 VECTOR QUANTIZED SEMANTIC EXTRACTION

Figure 9 highlights the crucial role of vector quantization (VQ) semantic representation extraction in our Mind4D framework. Without VQ, the high-level representation $z_{h,fg}$ results in ineffective image generation, which has cosine similarity of only 0.073, caused by high variation with fMRI and data scarcity. In comparison, our designed foreground and background extractors capture the semantic information, with an increased cosine similarity of 0.789, facilitating accurate reproduction of 4D scenes.

We further conduct empirical test on entropy and KL divergence as empirical validation. As shown in Table 4, our VQ design significantly lowers entropy and KL divergence by four to five orders of magnitude, which is consistent with our theoretical results.

Figure 10: **Ablation on fMRI input.** fMRI signals play a crucial role in 4D scene generation.

Table 4: Quantitative evaluation of the impact of vector quantization (VQ) design on entropy and KL divergence. Our VQ design reduces both measures by several orders of magnitude.

| VQ | Entropy (foreground) | Entropy (background) | KL (foreground) | KL (background) |
|---|---|---|---|---|
| w/o | $3.13 \times 10^5$ | $2.58 \times 10^5$ | $5.62 \times 10^2$ | $5.11 \times 10^2$ |
| w | **3.38** | **1.79** | $1.66 \times 10^{-2}$ | $9.58 \times 10^{-3}$ |

### D.5 Conditioning effect of fMRI signals

To examine the effect of fMRI input against prior knowledge from diffusion models, we performed an ablation on fMRI input. We isolate the impact of the fMRI signal by replacing the learned fMRI encoding with dimension-matched Gaussian noise while keeping every other diffusion model and the inference procedure identical. As shown in Figure 10, the whole scene is distorted without the fMRI signal as the condition. Although diffusion models can store and recover information, their output becomes noisy without meaningful fMRI signals. It illustrates that fMRI signals play a crucial role in 4D scene generation.

## E Further results on 4D generation

Figure 14 exhibits comparison of our method and other fMRI-based video reconstructions. Note that all previous works are only limited in 2D videos. In comparison, Mind4D proposed and realized the 4D generation with both spatial and temporal dimensions from brain signals. Additionally, Figure 12 shows the overrall 4D effects where dynamic images rendered from different viewpoints at different timestamps. Figure 11 shows more samples with subjects 1-3.

## F Proof of Theorem 3.1

In sparse sampling where the dimensionality of the decoded latent space $d = \dim(z_{\mathrm{h}})$ significantly exceeds the number of training samples $n$, that is $d \gg n$, the probability distribution $p(z_{\mathrm{h}})$ is not adequately represented. The empirical distribution $p(\hat{z}_{\mathrm{h}})$, which is approximated from a limited number of samples, fails to capture substantial portions of the probability mass inherent to $p(z_{\mathrm{h}})$.

For any $\delta > 0$, we consider a smooth-approximated empirical distribution encompassing a neighborhood with radius $r$: let $\hat{z}_{\mathrm{h}}$ be points in the decoded space such that $\|\hat{z}_{\mathrm{h}} - t_i\| > r$ for all $i \in \{1, \ldots, n\}$ with $t_i$ representing the training samples. For these points, it holds that $0 < p(\hat{z}_{\mathrm{h}}) < \delta$.

Denote $R_i$ as the union of all proximal areas around the training samples:

$$R_i = \bigcup_{i=1}^{n} U_i, \quad \text{where } U_i = \{u \in A : \|u - t_i\| \leq r\}, \tag{12}$$

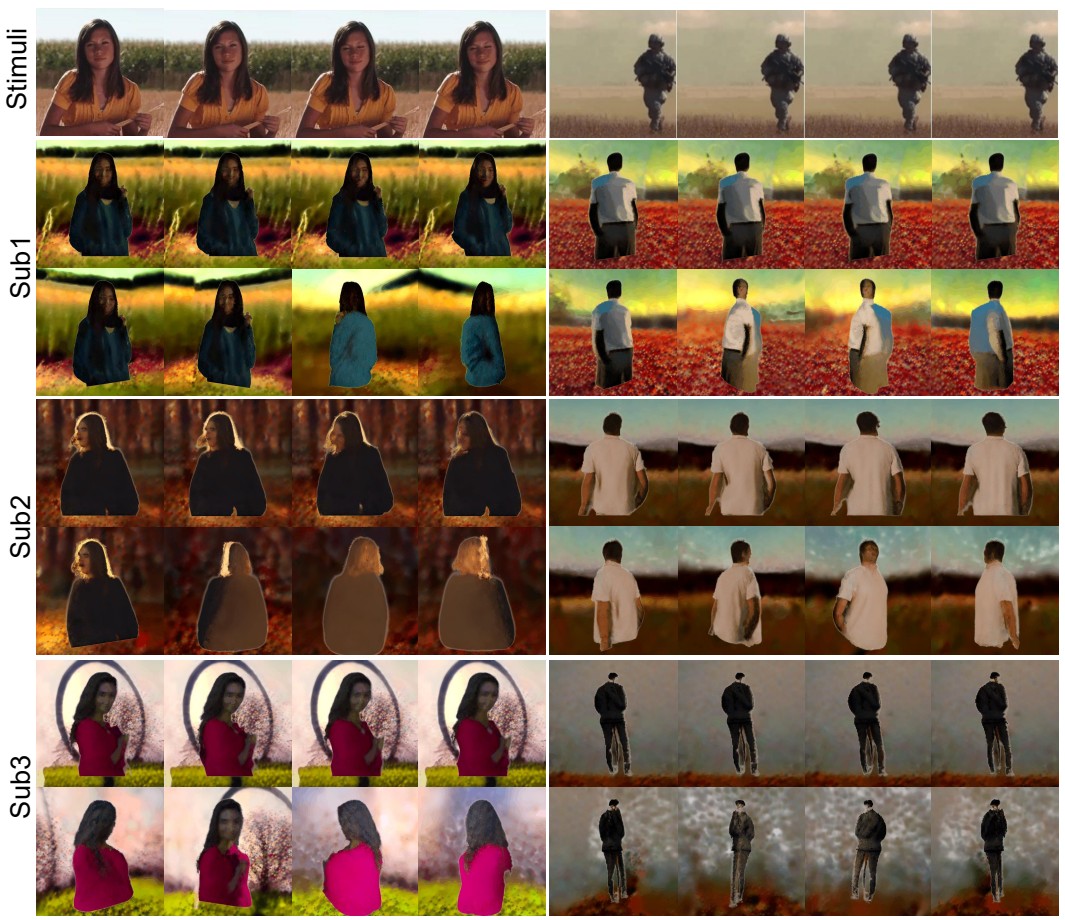

Figure 11: Samples from different subjects.

and let $R_o$ represent the complement region in the latent space $A$, which is far from the training samples:

$$R_o = A \setminus R_i. \tag{13}$$

Then the KL divergence without Vector Quantization will become:

$$
\begin{aligned}
& KL(p(z_{\mathrm{h}})||p(\hat{z}_{\mathrm{h}})) \\
& = \int p(z_{\mathrm{h}}) \log \frac{p(z_{\mathrm{h}})}{p(\hat{z}_{\mathrm{h}})} dz_{\mathrm{h}} \\
& = \int p(z_{\mathrm{h}}) \log p(z_{\mathrm{h}}) dz_{\mathrm{h}} - \int_{R_i} p(z_{\mathrm{h}}) \log p(\hat{z}_{\mathrm{h}}) dz_{\mathrm{h}} \\
& \quad - \int_{R_o} p(z_{\mathrm{h}}) \log p(\hat{z}_{\mathrm{h}}) dz_{\mathrm{h}} \\
& \geq \int p(z_{\mathrm{h}}) \log p(z_{\mathrm{h}}) dz_{\mathrm{h}} - \int_{R_i} p(z_{\mathrm{h}}) \log p(\hat{z}_{\mathrm{h}}) dz_{\mathrm{h}} \\
& \quad - \int_{R_o} p(z_{\mathrm{h}}) dz_{\mathrm{h}} \cdot \log(\delta) \\
& = O(\log \frac{1}{\delta}),
\end{aligned} \tag{14}
$$

which is relatively large when $\delta \to 0$.

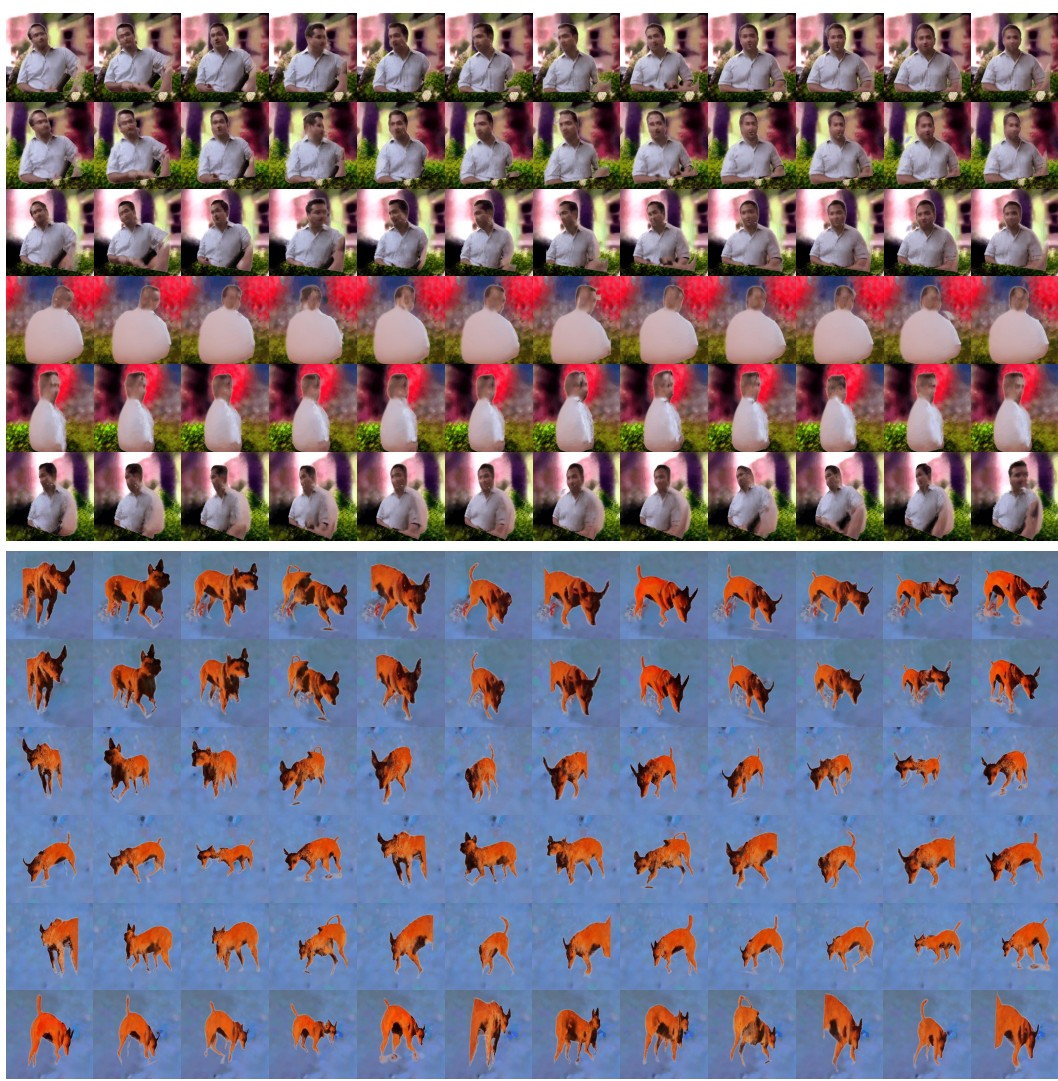

Figure 12: **4D results of two cases.** For each case, we show 6 viewpoints and 12 consecutive frames.

In an ideal scenario where the dataset is sufficiently large and evenly distributed, the region $R_o$ diminishes, effectively becoming negligible. Consequently, we could expect that:

$$KL(p(z_\mathrm{h}) \,\|\, p(\hat{z}_\mathrm{h})) = O(1), \tag{15}$$

as $R_o \to 0$. Conversely, in our setting where fMRI samples are sparse ($n \ll d$), a substantial region of $R_o$ persists, indicating a significant divergence in the decoded latent space.

After vector quantization, the number of samples $n$ greatly exceeds the number of quantization bins $K$. Assuming there is no disproportionate concentration of probability mass within these bins, the KL divergence becomes:

$$KL(p(z_\mathrm{h}^{vq}) \,\|\, p(\hat{z}_\mathrm{h}^{vq})) = \sum_{k=1}^{K} p(z_\mathrm{h}^{vq}) \log \frac{p(z_\mathrm{h}^{vq})}{p(\hat{z}_\mathrm{h}^{vq})} = O(1). \tag{16}$$

As a result,

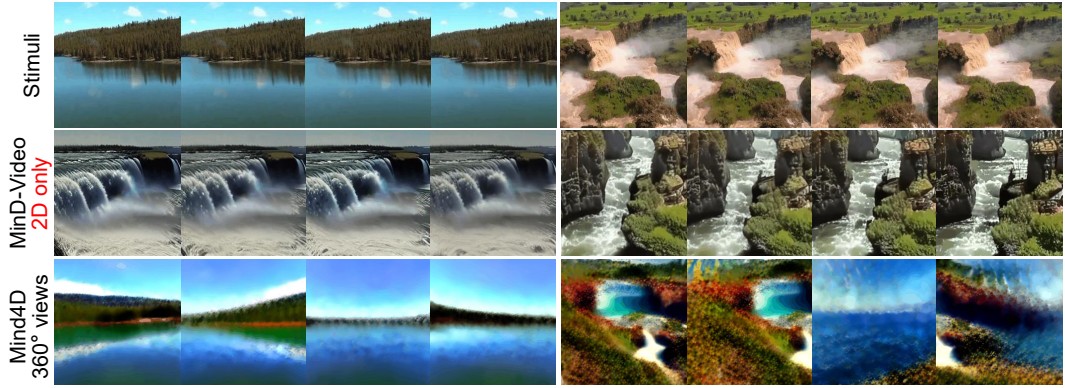

Figure 13: In background cases, Mind4D not only achieves consistent 360° rendering, but also delivers higher semantic accuracy with respect to ground truth stimulus.

$$KL(p(z_{\mathrm{h}}^{vq}) \,||\, p(\hat{z}_{\mathrm{h}}^{vq})) \ll KL(p(z_{\mathrm{h}}) \,||\, p(\hat{z}_{\mathrm{h}})). \tag{17}$$

## G  PROOF OF THEOREM 3.2

Assume that the high-dimensional latent space $A$ for $z_{\mathrm{h}}$ is confined within a closed hyperrectangle $[a_1, b_1] \times [a_2, b_2] \times \ldots \times [a_n, b_n]$ for each dimension. In a pretrained CLIP space as described in Radford et al. (2021), these bounds can be set to the extremal values obtained from decoding all pretraining images or texts.

Given any $\epsilon > 0$, one can choose a $\delta > 0$ such that $A$ is divided into a grid of smaller hyperrectangles. Specifically, we define a partition $(P_1, \ldots, P_d)$ where $P_i = (a_i = t_0 < t_1 < \ldots < t_{N_k} = b_i)$ with each interval $t_{j+1} - t_j$ being uniform and not exceeding $\delta$. Consequently, each subrectangle $S = [a_1', b_1'] \times [a_2', b_2'] \times \ldots \times [a_d', b_d']$ shares the similar volume $\Delta V_S$ and accommodates a integrated probability $\int_{S_j} P(z_{\mathrm{h}}) \, dz_{\mathrm{h}} = P(e_j)$.

Under the vector quantized extractor and for sufficiently small $\delta$, the quantized space can be further partitioned such that $P(e_k) = \sum_{j=1}^{J_k} P(e_{k_j})$, where $P(e_{k_j})$ represents the probability mass within the $j$-th partition of the $k$-th quantized space.

For each subrectangle $S = [a_1', b_1'] \times [a_2', b_2'] \times \cdots \times [a_d', b_d']$ of $P$ define its volume and bounds as:

$$v(S) = \prod_{i=1}^{d} (b_i' - a_i'), \tag{18}$$

$$m_S(f) = \inf f(x) : x \in S, \tag{19}$$

$$M_S(f) = \sup f(x) : x \in S. \tag{20}$$

Lower and Upper Riemann sums corresponding to the partition $P$ are then defined to be:

$$L(f, P) = \sum_{S \in P} m_S(f) \cdot v(S), \tag{21}$$

$$U(f, P) = \sum_{S \in P} M_S(f) \cdot v(S). \tag{22}$$

By the properties of Riemann integration, given any partition $P$ with norm $\|P\| < \delta$, it follows that:

$$U(f, P) - L(f, P) < \epsilon. \tag{23}$$

For each subrectangle $S$, we approximate the integrated probability over $S$ by selecting the 'average' value within this region, which is given by $\frac{P(e_k)}{\Delta V_S}$ and lies between $m_S(f)$ and $M_S(f)$.

$$L(f, P) \leq \int_S f(z_{\mathrm{h}})dz_{\mathrm{h}} \leq U(f, P), \tag{24}$$

$$L(f, P) \leq \sum_{i_1=1}^{N_1} \cdots \sum_{i_n=1}^{N_n} \frac{P(e_k)}{\Delta V_S} \log \frac{P(e_k)}{\Delta V_S} \Delta V_S \leq U(f, P). \tag{25}$$

Therefore, we have:

$$\sum_{i_1=1}^{N_1} \cdots \sum_{i_n=1}^{N_n} \left( P(e_k) \log \frac{P(e_k)}{\Delta V_S} - \epsilon \right) \tag{26}$$

$$\leq \int_{z_{\mathrm{h}}} P(z_{\mathrm{h}}) \log P(z_{\mathrm{h}})dz_{\mathrm{h}},$$

$$\int_{z_{\mathrm{h}}} P(z_{\mathrm{h}}) \log P(z_{\mathrm{h}})dz_{\mathrm{h}} \tag{27}$$

$$\leq \sum_{i_1=1}^{N_1} \cdots \sum_{i_n=1}^{N_n} \left( P(e_k) \log \frac{P(e_k)}{\Delta V_S} + \epsilon \right). \tag{28}$$

Consequently,

$$\lim_{\epsilon \to 0} H(z_{\mathrm{h}}, \epsilon) = -\sum_{i_1=1}^{N_1} \cdots \sum_{i_n=1}^{N_n} \left( P(e_k) \log \frac{P(e_k)}{\Delta V_S} \right). \tag{29}$$

As we consider the limit where $\epsilon \to 0$, it becomes feasible to represent the partitions of $A$ through their discrete counterparts.

We denote $H(z_{\mathrm{h}}) = \lim_{\epsilon \to 0} H(z_{\mathrm{h}}, \epsilon)$ as the entropy pf Riemann-Discrete approximated distribution of the embeddings after MLP $z_{\mathrm{h}} = f_e(X)$ without vector quantization. Then, we have:

$$H(z_{\mathrm{h}}) = -\sum_{k=1}^{K} \sum_{j=1}^{J_k} P(e_{k_j}) \log \frac{P(e_{k_j})}{\Delta V_S}. \tag{30}$$

$$H(z_{\mathrm{h}}^{vq}) = -\sum_{k=1}^{K} P(e_k) \log P(e_k) \tag{31}$$

$$= -\sum_{k=1}^{K} \sum_{j=1}^{J_k} P(e_{k_j}) \log P(e_k). \tag{32}$$

We operate under the hypothesis that the probability distribution is dispersed across the space, which precludes significant localization or the emergence of regions with disproportionately high probability mass. This is a plausible assumption within a space that has been pretrained with a large set of data, thereby approximating a well-spread distribution. Formally, we can express this as

$$J_k = O\left( \frac{L^d}{K \Delta V_S} \right), \text{ or to say } J_k = c_k \frac{L^d}{K \Delta V_S}. \tag{33}$$

where $c_k$ is a constant of order 1 ($c_k = O(1)$) and strictly positive ($c_k > 0$). In the case where the scale of the space $L$ is large and the dimensionality $d$ is much larger than the number of quantization bins $K$, the ratio $\frac{K}{L^d}$ becomes vanishingly small, implying that $c_k \ll \frac{K}{L^d}$, leading to the result:

$$P(e_k) = O\left(\left(\frac{L^d}{K\Delta V_S}\right)P(e_{k_j})\right), P(e_k) > \frac{P(e_{k_j})}{\Delta V_S}. \tag{34}$$

The implication here is that the entropy of the decoded space $H(z_{\mathrm{h}})$ is greater than that of the quantized space $H(z_{\mathrm{h}}^{vq})$, accounting for the additional logarithmic factor:

$$H(z_{\mathrm{h}}) - H(z_{\mathrm{h}}^{vq}) = O\left(\log\left(\frac{L^d}{K}\right)\right), H(z_{\mathrm{h}}) > H(z_{\mathrm{h}}^{vq}). \tag{35}$$

The difference $\log\left(\frac{L^d}{K}\right)$ particularly large in our specified setting when the dimensionality $d$ is much less than the number of fMRI samples $n$, which in turn is substantially less than the number of quantization bins $K$, and considering the large size of the CLIP space denoted by $L$.

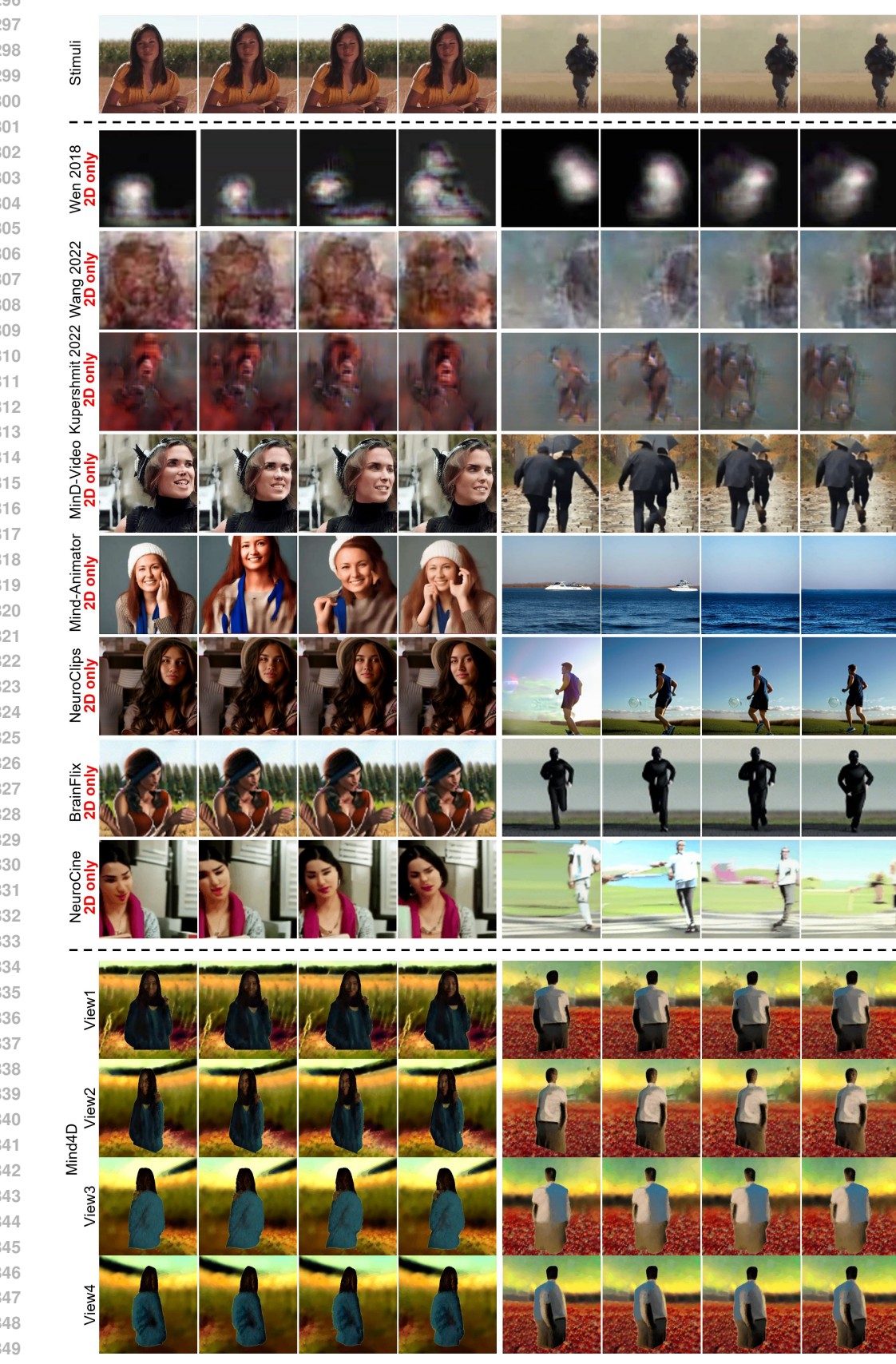

Figure 14: **Comparisons with more fMRI-based video generation.** All previous methods (Wen Wen et al. (2018), Wang Wang et al. (2022), MinD-Video (Chen et al., 2024), Mind-Animator (Lu et al., 2025)) are limited in **2D** when there is only 2D supervision. In comparison, Mind4D pinoeers the **Brain-to-4D** function through a novel weakly supervised framework. See the video in supplementary for dynamic results.

