# OpenReview forum: "4D Mind Reading"
_ICLR.cc/2026/Conference — Submitted to ICLR 2026_

### Official Review · Reviewer_1QE8 · 2025-10-27

**Soundness:** 2
**Presentation:** 3
**Contribution:** 2
**Rating:** 2
**Confidence:** 5

**Summary:**

The work introduces the novel and ambitious task of "4D Mind Reading," aiming to reconstruct dynamic 3D scenes from fMRI signals. The ambition to extend brain decoding beyond 2D video is commendable, and the technical integration of various state-of-the-art generative models is non-trivial. However, the paper's core premise raises fundamental questions about the scientific validity and utility of the proposed task. The method deviates from the primary goal of visual decoding—reconstructing the subject's perceptual experience—by generating novel 3D views that were never seen. Methodologically, the framework appears to be a two-step cascade: a standard fMRI-to-video decoder followed by a state-of-the-art video-to-4D generative model, which limits the novelty in the neural decoding aspect itself. The practical value and neuroscientific relevance of the generated, non-perceived 4D content are not convincingly justified, weakening the overall contribution.

**Strengths:**

1. Novel Task Formulation: The paper is the first to propose the concept of "4D Mind Reading," pushing the boundaries of neural decoding from reconstructing 2D images/videos to generating immersive 4D scenes. This direction is conceptually novel and ambitious.

2. Sophisticated Technical Framework: The proposed Mind4D framework is technically sound in its design, cleverly leveraging existing fMRI-2D video datasets through a weakly supervised approach. The decomposition of scenes into foreground and background and the integration with powerful generative priors like 4D Gaussian Splatting represent a sophisticated engineering effort.

**Weaknesses:**

1. The fundamental premise of the task is debatable. The goal of visual decoding is typically to reconstruct what a person is seeing or imagining. Generating novel 3D views that were never part of the visual stimulus (and thus whose specific geometric information is not directly encoded in the fMRI signal) is not "decoding" but rather "generation conditioned on a decoded signal." The authors fail to adequately justify why this "4D hallucination" is a meaningful or scientifically valid objective for a BCI, especially when the generated content does not reflect the subject's actual perception.

2. The framework's contribution to the neuroscience problem of brain decoding is incremental. The pipeline effectively separates into two distinct parts: (1) an fMRI-to-2D-representation decoding stage and (2) a representation-to-4D-scene generation stage. The heavy lifting for the "4D" aspect is performed by pre-trained generative models in the second stage, which operate independently of the brain signal's constraints. The paper feels more like a new application for 4D generative models rather than a significant breakthrough in understanding brain representations.

3. The authors acknowledge the absence of 4D ground truth, which severely hampers meaningful evaluation. The used metrics (CLIP-T-G, user study on geometry) are indirect, subjective, and primarily assess the quality and consistency of the 4D generation, not the accuracy of the 4D decoding. It is impossible to know if the generated 3D geometry bears any resemblance to the actual 3D scene from which the stimulus video was originally filmed.

4. The claims regarding applications in "neuroscience interpretability and clinical diagnosis" are highly speculative and not convincingly supported. For example, the ROI analysis (Figure 5) is performed on the basis of a synthetically generated 4D scene, which is several steps removed from the actual brain activity. Drawing neuroscientific conclusions from how different brain regions contribute to a "hallucinated" 3D structure could be fundamentally misleading.

5. The pipeline is exceedingly complex, relying on a cascade of multiple large pre-trained models (CLIP, VAE, SDXL, SAM, various 3D/4D priors). The performance is likely highly sensitive to the specific choice and quality of these models, and the framework's generalization to different fMRI datasets, acquisition parameters, or types of visual stimuli remains unproven. The extensive engineering makes it difficult to pinpoint the core scientific insights.

**Questions:**

nan

---

### Official Review · Reviewer_shc5 · 2025-10-28

**Soundness:** 1
**Presentation:** 3
**Contribution:** 2
**Rating:** 2
**Confidence:** 4

**Summary:**

The paper's main contribution is its novel objective to reconstruct dynamic 3D scenes (4D) from fMRI signals, moving beyond existing 2D video or static 3D methods. It proposes Mind4D, a weakly-supervised framework designed to function without direct fMRI-4D training data. The model's architecture uses a hierarchical system to decode separate semantic and detail representations from fMRI. The authors report that 2D projections of their generated 4D scenes achieve higher scores on specific semantic benchmarks compared to baseline 2D-only methods.

**Strengths:**

The strengths of this paper lie in its presentation, which includes rich background studies, clear figures, theoretical analysis as well as quantitative studies. The idea of generating 4D scenes is interesting.

**Weaknesses:**

- The method assumes that the brain signals from watching a 2D video are a sufficient proxy for 4D perception. However, the brain processes 2D projections very differently than it processes immersive 3D environments (which involve depth, parallax, and spatial navigation). The model may not be "reading" 4D perception at all; it might just be learning to "hallucinate" a plausible 3D shape from signals encoded from a 2D video.

- The model is forced to find fMRI patterns that correlate with an external, human-provided 2D label. Is there evidence that the brain separates information this cleanly or in the same way?

- The qualitative results (Fig. 4) show multi-view consistency (i.e., the generated object looks 3D and stable when rotated) but not multi-view accuracy (i.e., is it the correct 3D shape?). The model could be consistently generating a semantically plausible but entirely incorrect 3D structure, and the paper's evaluation metrics would not catch this. Lacking a proper groundtruth would be a severe issue.

- The paper claims to outperform state-of-the-art 2D video generation models (like MinD-Video) by "flattening" its 4D output into a 2D video. This is an apples-to-oranges comparison. The Mind4D model was given extra, privileged information that the 2D baselines were not: the manual foreground/background annotations. This additional supervision forces the model to learn a better scene decomposition, which naturally leads to better semantic scores (like ICS-50). The improvement might not come from the 4D modeling itself, but simply from the extra-labeled data.

**Questions:**

The questions are included in the weaknesses. I don't have extra questions.

---

### Official Review · Reviewer_LUuy · 2025-10-29

**Soundness:** 2
**Presentation:** 2
**Contribution:** 2
**Rating:** 2
**Confidence:** 4

**Summary:**

This paper proposes Mind4D, a novel brain-inspired fMRI-conditioned 4D generation framework, which generates dynamic 3D scenes with spatiotemporal 4D visuals directly from fMRI signals. The authors argue that previous research on fMRI-to-image, video, or 3D shape reconstruction fails to capture the full spatiotemporal perception of human cognition. Mind4D employs a weakly supervised asymmetric hierarchical representation learning approach, leveraging existing fMRI–2D datasets to derive hierarchical representations that condition foreground and background generative priors based on diffusion and Gaussian splatting. Experimental results show substantial improvement in ICS-50 over state-of-the-art fMRI-to-video baselines.

**Strengths:**

1. The decomposition of foreground and background for separate decoding is conceptually insightful.
2. The supplementary material provides clear reconstruction examples that effectively demonstrate the results.
3. The ablation studies are comprehensive and cover the core contributions of the method.

**Weaknesses:**

**Motivation**

1. The paper argues that brain-to-4D reconstruction provides a better “user experience” and more faithfully reflects “human perception” than image or video reconstruction. However, such advantages are not clearly demonstrated in the demos. Given the current signal-to-noise ratio of neuroimaging data and the limitations of existing neural decoding algorithms, the brain-to-4D task appears overly ambitious at this stage.
2. During fMRI acquisition, the subjects were exposed only to 2D visual stimuli, raising a fundamental question of how 4D information could be encoded in the brain signals. The reconstructed 4D structures may largely arise from the model’s generative priors rather than genuine neural representations, making the conceptual significance of brain-to-4D reconstruction unclear.

**High-level / Low-level / Encoding**

1. The interpretation of **"high-level"** and **"low-level"** features is questionable. While CLIP embeddings can be considered high-level, VAE features cannot reasonably be regarded as low-level representations. Moreover, using the original image for “high-level” encoding and a Gaussian-blurred image for “low-level” encoding makes no sense.
2. The motivation for introducing fMRI encoding is insufficient, and the design raises major concerns. What is the purpose of encoding? Is it merely for data augmentation through extra images? Encoding directly from images is suboptimal. "High-level" and "low-level" encoding are entirely different concepts from the "high-level" and "low-level" features in decoding. How does this enhance decoding? For these reasons, the effectiveness of encoding and the quality of the augmented data are doubted.
3. Are the modules (encoder, decoder, extractor) trained simultaneously or separately? There is a lack of individual result reports for each module.

**Foreground / Background / Generation**

1. Why are both the foreground and background extracted only from the high-level features? How exactly are the supervision signals (codebook) for the foreground and background obtained? The paper only seems to mention "through our prelabeled foreground and background annotation" (line 218).
2. The supervision signals for both the foreground and background originate from the CLIP features of the 2D video, with no supplementary 4D information? Can it be concluded that the foreground and background features are not decoded from fMRI, but entirely derived from the generative model's prior? Considering the motivation mentioned—that the stimuli seen by the subjects are 2D videos—this task seems more like a combination of brain-to-2D and 2D-to-4D, rather than brain-to-4D.
3. In Fig. 3, the 2D background images are primarily generated by $z_h$ through SDXL. Although $z^{vq}_{h,bg}$ is introduced via SAM and inpainting, its role is minimal. This raises a question: since the decoding of $z_h$ is supervised by the CLIP features of the 2D video rather than the background features, how can it reconstruct the correct background?

**Questions:**

see Weaknesses

---

### Official Review · Reviewer_gsjT · 2025-11-01

**Soundness:** 2
**Presentation:** 3
**Contribution:** 2
**Rating:** 2
**Confidence:** 4

**Summary:**

The paper proposes a brain-conditioned pipeline that maps fMRI to hierarchical latents (a high-level CLIP-aligned code and a low-level VAE-like code), and then drives multiple powerful vision priors (diffusion/video/3D-GS) to synthesize dynamic 3D (“4D”) scenes from brain activity. Training mixes paired fMRI–video with surrogate fMRI generated from images. Experiments are on Wen-style video-fMRI data with 2D and “4D” metrics.

The idea is ambitious and the visual results are engaging. However, the current experimental design does not isolate how much of the output is truly driven by fMRI versus by the strong generative priors; and the “4D” evaluation lacks reliable ground truth or geometry-aware checks. Claims about correspondence to human brain mechanisms also outpace the evidence.

**Strengths:**

## Strengths

* Interesting regularization idea: Using VQ to coarsen CLIP-aligned targets is a sensible way to curb overfitting; the mathematical discussion provides useful intuition to the formulation.
* Data augmentation concept: Training an image→fMRI encoder and using it to generate fMRI for additional images is a creative way to expand supervision; ablations suggest it helps downstream recon quality.

**Weaknesses:**

### 1. Methodological side (what is fMRI-driven? It's not related to this work specifically, but this line of work in general)

* Similar to previous work, the pipeline leans heavily on **strong priors** (e.g. SD/diffusion, video priors, Zero-NVS/MVDream, SAM/inpainting). Without comprehensive **brain-signal attribution tests**, it’s unclear whether improvements reflect genuine brain decoding or prior-driven hallucination. The current “fMRI→noise” visual control is insufficient and not quantified.
* The surrogate-fMRI augmentation risks **self-confirmation**: encoders trained on the same dataset then generate labels that the decoder learns to invert, potentially amplifying vision priors rather than brain alignment.

### 2. Neuroscience framing feels loose

* The claim that the architecture reflects “dual functions of the human brain” (environment decoding + representation encoding) feels neuro-theoretically loose. The brain plausibly performs hierarchical encoding of sensory inputs leading to higher-level semantics; but the paper doesn’t present encoding/decoding analyses showing selective mapping of learned latents to ROIs beyond simpler baselines.

### 3. Task feasibility and evaluation of “4D Mind Reading” (important point)

* On Wen dataset, we have **front-view ground truth only**; other views are **model-generated**. If multi-view supervision is synthetic, the “4D mind reading” objective can become circular (matching the generator’s own biases). The paper needs **geometry-aware, view-consistency** metrics and non-synthetic checks.
* Reported Table 1 results over multiple views lack clarity on how “ground truth” for non-front views is defined, and how baseline numbers were obtained. This raises fairness concerns.


### 4. Encoding design

* Mapping both **high-level semantic images** and **low-level images** to the same fMRI target is questionable without task-specific gating or an explanation of how the model resolves the semantic mismatch. This can conflate distinct representational levels and encourage shortcutting through priors.

**Questions:**

1. Attribution: Provide **quantitative** brain-signal controls:
   * maybe shuffle fMRI labels within batches as one of the control; time-reverse or phase-randomize fMRI; cross-subject input; progressively mix fMRI with noise. Report Δ on your core metrics.
2. Ground truth & fairness: Please precisely define the supervision and evaluation for non-front views; re-report baselines on the **same** front-view ground truth and also on standard 2D recon benchmarks. Explain any discrepancies with published numbers.
3. Some suggestions:
* **Ablate priors**: remove Zero-NVS backgrounds; freeze or randomize BLIP prompts; replace SD keyframes with blurred frames; quantify drops to expose which gains are brain-driven.

## Minor Issues

* Line-level typos you flagged (L190 extra period; L208–209 unclear expression) and the Fig. 14 caption overflow should be fixed.

---

### Meta-Review · Area_Chair_aTD1 · 2026-01-06

**Summary:**

The submission tackles an ambitious “4D mind reading” setting and proposes Mind4D, combining hierarchical decoding from fMRI with powerful generative priors to produce dynamic 3D reconstructions. While the visual results are appealing, the current evidence does not convincingly establish that the reconstructions are driven by neural signals rather than largely by the model prior, and the absence of reliable 4D/geometry ground truth makes it difficult to substantiate the central claims quantitatively. Overall, the work is promising in direction but falls short of the scientific rigor expected for acceptance at this stage.

**Reviewer Concerns:**

Across reviews, the main blockers are methodological rather than expository. First, causal attribution remains unclear: the paper would benefit from stronger negative controls and stress tests (e.g., fMRI shuffling, controlled noise injection, and targeted ablations) that quantify how sensitive the outputs are to the actual brain signal versus the prior. Second, evaluation is not yet convincing: without objective 4D/geometry ground truth, reported metrics can become indirect and potentially circular, especially when large parts of the scene are effectively synthesized by the generative pipeline. Third, comparisons and framing raise concerns: the use of additional supervision (e.g., manual annotations) can confound comparisons to 2D baselines, and the neuroscience narrative appears stronger than what is currently supported by controlled evidence. Since there was no author rebuttal, these points were not clarified or resolved during the discussion period.

**Reviewer Scores:**

All four reviewers gave a 2 (Reject) with high confidence, and their critiques were notably consistent in focusing on validity, controls, and evaluation design. In the absence of an author response, there was no new information that would plausibly shift these assessments, so the scores would be expected to remain unchanged.

---

### Decision · Program_Chairs · 2026-01-26

Reject